# Emergencies in Long-Term Care Services for the Elderly in Korea: A Mixed-Methods Study

**DOI:** 10.3390/ijerph17010066

**Published:** 2019-12-20

**Authors:** Soon Ok Kim, Sun Hee Bae

**Affiliations:** 1Department of Nursing, Shinhan University, Uijeongbu 11340, Korea; kso6210@hanmail.net; 2Department of Nursing, Kyungbok University, Pocheon 11138, Korea

**Keywords:** aged, long-term care, emergencies, Korea

## Abstract

This mixed-methods study explored ways to enhance the emergency response abilities of workers in long-term care services for the elderly. Based on different service types, we identify emergency situations and the response abilities of workers in long-term care services. Results indicated that there are more emergency situations in care facilities than in home care services. However, 71.3% of respondents in facilities said emergency response abilities were low compared to 44.2% of workers in home care services. Qualitative research identified six categories and 16 themes based on emotions experienced during emergencies and the challenges in determining solutions. The study confirms that there is a difference in emergency incidences and the coping abilities of workers in facilities and home services with high emergency incidence rates. Developing and applying guidelines for emergency response management by service type is recommended.

## 1. Introduction

In 2015, over 126 million people were aged 80 years or above, representing a 57.2% increase in the number of elderly people since 2000. This number is expected to triple by 2050 [1], indicating that the aging population has become a critical issue worldwide. Korea is the fastest aging society globally and it officially became an aged society in 2017. By 2025, Korea is expected to become a super-aged society, with more than 20% of the population being over the age of 65 years. Korea adopted a long-term health care insurance system in 2008 to address this challenge [2]. This insurance system provides long-term care services (LTCS), such as physical activities and housekeeping support, for the elderly who are no longer independent owing to dementia or physical disease. Two branches of LTCS include facility care services (FCS), which are provided when an elderly patient is admitted to a medical welfare center, and home care services (HCS), which are provided when an elderly patient requires a long-term caregiver in their home [3].

Recipients of LTCS include elderly patients with dementia, Parkinson’s disease, and cerebrovascular diseases [4]. Patients have typically suffered from these diseases for a long time and are vulnerable to further complications due to their chronic diseases combining with various other complexities from acute conditions (e.g., urinary tract infection, pneumonia, dehydration), which can lead to emergencies that require immediate treatment [5]. LTCS classify the elderly into five grades based on their health conditions or physical dependence. The choice of FCS or HCS provision is based on this grade. FCS are for people with severe physical dependence (Grades 1 and 2), and HCS are reserved for those with reduced physical dependence (Grades 3–5). Thus, those receiving FCS are likely to have complex diseases, weakened immune systems, and restricted mobility, which increases their risk for bedsores, infections, and further physical dependence [6].

More than 75% of emergencies involving the elderly occur in their homes, with 45.8% being fall-related and 9.7% involving asphyxia. Elderly people over 65 years account for 66.6% of all asphyxia-related deaths [7]. Emergencies involving the elderly in LTCS sites are frequent regardless of service type, which highlights the importance of developing methods that can accurately identify abnormal changes in patients and quickly recognize emergencies to prevent accidents and save lives. Additionally, appropriate responses to emergencies by practitioners of LTCS are crucial [8] and need to be performed quickly and accurately during this critical window. Elderly patients require a longer recovery period from deterioration or injury, and the risk of secondary complications during this process is high if not addressed accordingly from the onset. Moreover, it is usually elderly people with severe, difficult to manage illnesses that are FCS patients, which results in lower daily living performance [9], more exposure to emergencies, more frequent use of emergency departments [10], longer hospital stays, and increased mortality rates when compared to HCS patients [11].

Conversely, HCS patients have less severe illnesses, yet the proportion of HCS is twice as high (48.5%) as the proportion of FCS (21.9%) [12]. These values indicate that it is necessary to apply different emergency responses according to the type of care service and the type of emergency that is most likely to occur. Despite the need for improvement of long-term care workers’ ability to respond to emergencies, no specific information has been gathered regarding the type and frequency of emergencies at different service sites, nor has information been gathered regarding how they are perceived, and what the current response capacity is. 

As far as we are aware, no studies have identified the emergency situations and response abilities in LTCS. Consequently, we aim to identify and analyze these variables using a mixed-methods design that measures the types of emergencies involving the elderly and the ability of workers to respond to these emergencies. This study is not limited to identifying various emergencies through quantitative research, but also provides an in-depth look at the experiences of research participants during emergencies. In order to analyze emergency situations systematically, a mixed study was conducted. Moreover, we extended our mixed-methods design to include focus group interviews that explored the meaning, content, and nature of workers’ live experiences regarding emergencies involving the elderly, which also helped identify their ability to respond to emergencies. 

### Study Objectives

The study aims to first identify and quantitatively analyze emergency situations involving the elderly and the required emergency response abilities of workers in LTCS. Second, it phenomenologically explores through focus group interviews with workers, the meaning and nature of live experiences regarding emergency situations involving the elderly and the necessary emergency response abilities.

## 2. Materials and Methods

### 2.1. Research Design and Procedure

Using a mixed-methods design, a descriptive survey (quantitative) and two focus group interviews (qualitative) were conducted to identify emergencies involving the elderly and the LTCS workers’ abilities to respond to these emergencies. Using this design, we were also able to explore the meaning, content, and nature of workers’ live experiences regarding emergencies involving the elderly and their perceptions of their ability to respond to these emergencies.

### 2.2. Study Participants

Participants for the descriptive survey included directors of elderly care facilities (ECF) and home care centers (HCC) who understood the purpose of the study and provided written consent. In Korea, LTCS for seniors are divided into ECF and HCC. For the first time, this study involved the directors who are responsible for ECF and HCC to accurately grasp the emergency situations occurring in both specialized facilities and home care services for the elderly.

A power analysis was conducted using G*Power 3.1, with the significance level α = 0.05, statistical power 0.95, and an effect size of 0.5. This analysis estimated that a sample size of 176 was needed. Excluding nine incomplete surveys, 259 surveys were used for the final analysis. Participants for the focus group interviews consisted of nine women, comprising six at an ECF and three at HCC, who met the inclusion criteria of being able to communicate clearly, with consistent thinking and judgment, understand the purpose of the study, and provide written consent.

### 2.3. Data Collection

Data were collected between 1 to 28 February 2019. For the quantitative research, the survey was distributed to workers who wished to participate in the research at the Korean Senior Welfare Association and the Korean Association of the Korean Elderly Welfare Association. 

For the qualitative study, two focus group interviews were conducted in a conference room and lasted approximately two and a half hours. The specific selection criteria were those who understood the purpose of the study and gave written consent to participate, among those who were able to communicate, had a clear flow of thought and judgment, and were consistent in their opinions.

A voice recorder was used with participants’ approval. The main interview questions were, “What was the emergency like when you were taking care of the elderly?” and “How did the ECF and HCC cope with the emergency?”

## 3. Measurements

### 3.1. Quantitative Research

To identify emergencies involving the elderly and the ability of workers to respond to these emergencies, we modified and supplemented tools developed by Lee [6] before transporting patients to the emergency room. The use of these tools was to investigate how best to respond to, and transport, emergency elderly patients. Three nursing professors, three ECF directors, and two HCC directors reviewed the survey before we made final revisions, and they approved its use. The content validity index (CVI) was found to be above 0.82. The final questionnaire survey examined 19 items of emergency experience, 19 items of first aid, five items of emergency response abilities, and six general characteristics.

Experiences of emergency situations were measured on a 4-point Likert scale. 1 point was given for fewer than 5 emergency situations experienced in a year; 2 points for 6–10 situations; 3 points for 11–20; and 4 points for 21 times or more. Higher scores indicated more first-aid experience. Internal consistency for emergency experience used Cronbach’s α = 0.76, as described by Lee [6]. Cronbach’s α for this study’s internal consistency was 0.88. For the ability to respond to emergencies, Cronbach’s α = 0.80, as described by Lee [6]; however, Cronbach’s α for this study’s ability to respond to emergencies was 0.85, and 0.78 for first-aid. 

### 3.2. Statistical Analysis

Quantitative research was analyzed using the SPSS WIN 22.0 program. Participants’ demographic characteristics, number of emergencies, type of emergencies, and the ability to respond to emergencies involving the elderly were analyzed using frequency data. An independent-samples t-test was performed to compare the number of emergencies experienced and first-aid by service type, while a chi-square test measured workers’ ability to respond to emergencies. An independent-samples *t*-test and one-way ANOVA were performed to compare the differences in the participants’ demographic characteristics. 

The Colaizzi analysis procedure [13] was used for qualitative analysis. This procedure is a phenomenological approach that enabled participants to find clear meaning from the content described and accurately state the nature of the phenomenon. The reliability and validity of the characteristics and content identified were evaluated through a consultation with a professor of geriatric nursing and a professor of emergency nursing. 

### 3.3. Ethical Considerations

To protect our participants, ethical approval was acquired from the research ethics committee (IRB) of the researchers’ respective universities (Approval number: SHIRB-201806-HR-084-02).

## 4. Results

### 4.1. Quantitative Results

#### 4.1.1. Participants’ General Characteristics

The demographic characteristics of the participants are shown in Table 1. In HCC, 89.9% of directors were social workers, and 84.6% were social workers at ECF. In an emergency, 33.6% of caregivers in HCC first reported to the 119 rescue team, whereas 43.8% of ECF reported to nurses. In HCC, 48.1% of those in charge of first aid were caregivers, and 21.3% were facility director; in ECF, 33.1% were nurses, and 17.7% were facility director. When transferring patients to a hospital, HCC caregivers accompanied the patient 47.5% of the time, and social workers did so 14.9% of the time. In ECF, the numbers were 45.4% for caregivers and 23.2% for nurse’s aides. The emergency experience was 100% in both the ECF and HCC.

#### 4.1.2. Emergency Type and First-Aid

In all items except hypoglycemia, the ECF participants experienced more emergencies than the HCC participants, and there was a significant difference between the two groups. However, intoxication was not significantly different between the two groups (Table 2).

Regarding first-aid by service type, the average for ECF was 0.60 ± 0.21 points, and HCC was 0.42 ± 0.20 points, which is a significant statistical difference (*p* < 0.001). The order of ECF first-aid was (1) call patient’s guardian, (2) check vital signs, and (3) call 119; whereas the order of HCC first-aid was (1) call 119, (2) call patient’s guardian, and (3) check vital signs. There was a significant difference in all treatment items except assistance with medication, blood glucose test, the Heimlich maneuver, and calling 119 (Table 2).

#### 4.1.3. Emergency Response Abilities

Participants’ self-reported ability to respond to emergencies were “low” (ECF = 47.7%; HCC = 34.9%; χ^2^ = 27.115, *p* < 0.001), and 98.5% of ECF and 72.1% of HCC answered “yes” to having experience of transportation to hospital (χ^2^ = 35.961, *p* < 0.001). The number of transfers was below 5 cases (ECF = 45.4%; HCC = 77.5%; χ^2^ = 47.966, *p* < 0.001). The decision to transfer a patient to the emergency room was discussed between the facility director and the patient’s guardian (ECF = 34.6%; HCC = 33.3%), which showed a statistically significant difference (χ^2^ = 43.716, *p* < 0.001). When asked if they had experience with the regulations for emergency procedures, 54.3% of HCC participants and 54.6% of ECF participants answered: “yes” (Table 3).

### 4.2. Qualitative Results

#### 4.2.1. Results of Qualitative Analysis

The director participants were all females aged between 43 and 61 years. Seven had a bachelor’s degree or above, and two had associate degrees. Six worked at ECF and three at HCC. All nine were facility directors. Working experience at LTCS ranged from 5 years and 8 months to 17 years and 3 months. 

#### 4.2.2. Elderly Care Facilities and Home Care Services Directors’ Experience of Emergencies and Their Ability to Respond to Emergencies

One hundred and six structured experiences were extracted from the original data. Furthermore, 44 themes were derived, and 16 theme clusters were drawn from them. Six categories were presented through systematic conceptualization, and included (1) confusion in recognizing emergencies, (2) sudden occurrence of an emergency, (3) unsystematic emergency training (emergency situation training that is not systematic), (4) non-independent response (i.e., decision made in consultation with others), (5) legal standards that do not fit reality, and (6) psychological withdrawal associated with death (Table 4).

#### 4.2.3. Description of the Meaning of ECF and HCC Directors’ Experience of Emergencies and Their Ability to Respond to Emergencies

Category 1. Confusion in recognizing emergencies

Participants were confused because they could not clearly identify emergencies in their facilities and had conflicting and contradictory concepts of death and emergencies.
“I don’t know exactly what to call an emergency, but since we’re not nurses...I look at it as an emergency if there’s something unusual”.(Participant 4)

Category 2. Sudden occurrence of an emergency

Participants experienced sudden emergencies due to hypoglycemic shocks and intrauterine bleeding, as well as airway obstructions from secretions in the airway.
“If there’s bleeding, we, of course, attempt to stop the bleeding. A frequent emergency is vaginal bleeding. There are quite a few instances in elderly people where they start bleeding while putting on diapers”.(Participant 2)

Category 3. Unsystematic emergency training

The ECF and HCC directors were disappointed about the lack of compulsory training when running the facility, especially because there has been no training for emergencies.
“Only 6% can self-train for emergencies. That’s because nurses are stationed at facilities. So, the remaining 94% are left unattended. So, we have to make it mandatory”.(Participant 1)

Category 4. Non-independent response

Participants spoke to the patient’s family before taking the patient to the hospital. By discussing with family members, they could take necessary action before the emergency occurred. Additionally, participants dealt with emergencies by assigning work to nurses.
“If they have a fever and are unconscious, we discuss with their family, if they have any, and take them to the hospital”.(Participant 6)
“In our case, the part-time doctor comes twice a week. He receives calls in emergencies, so you can get help right away”.(Participant 9)

Category 5. Legal standards that do not fit reality

Participants were confused regarding the standards for medical practice in ECF and HCC, as they had different response criteria, which were unclear.
“When an accident occurs in clinical work, it gets very difficult when the accident involves legal matters. Since they consider precedents and the law, most cases involving care homes are caught legally”.(Participant 6)

Category 6. Psychological withdrawal associated with death

Participants felt burdened by being investigated about a death immediately after the emergency and were frustrated by being questioned at the police station.
“As you said, when the person died, they came from four places. It was too complicated. They came from the police station, and the first person to make the discovery was really traumatized”. (Participant 3)
“When I went to the prosecutor’s office, they asked me, ‘Why did you let the patient die by not checking on them every three minutes? The prosecutor... That’s why I was so intimidated. For some reason, I felt like I had done something wrong”.(Participant 1)

## 5. Discussion

Our quantitative study’s results revealed that all of our participants have experienced an elderly patient-related emergency, with the ECF participants having experienced an average of 1.93 (6–10 experiences of emergency over a 1-year period) emergencies, and the HCC participants having experienced an average of 0.78 (experienced less than 5 emergency situations in 1 year) emergencies during the study period. Consequently, ECF participants experienced emergencies more often, and these results indicate that patients requiring FCS are at an increased risk for emergencies. These findings correlate with previous research [14], which indicated that the elderly in FCS were at an increased risk for emergencies due to the severity of their underlying conditions and deterioration, which could explain why fewer emergencies occurred in HCS compared to FCS in our study since these elderly patients had less severe conditions. Although fewer emergencies occur in HCS, some elderly patients may have high medical demands due to chronic diseases and limited mobility [15]; hence, patients with Grades 1–2 can benefit from services in both FCS and the HCS, and those requiring HCS may officially be rated as Grade 1 or 2. In this study, when an emergency occurs, the decision to transfer the patient to the emergency room is reached through consultation with their guardian or guardian at home. There was no difference between the two groups in this regard. This is because the guardian makes the decision to transfer to the emergency room regardless of the facility, because traditionally Korean society is centered on the family. In addition, workers were the most frequently used as companions when patients were transferred to the emergency room. As previous studies have shown, both FCS and HCS are the people who care for the elderly in the front line [6].

Despite the frequency of emergencies, 71.3% of participants in ECF (44.2% in HCC) said their ability to respond to emergencies was “low”. Interestingly, ECF participants said they felt less-equipped than HCC participants. In this study, most facility directors were social workers and not medical workers, which is in line with the 2014 findings that revealed that despite there being many severe patients in ECF, only 6% of the workers were nurses. In fact, only 33.1% of ECF emergencies had nurses responsible for first-aid, whereas the remaining institutions had non-medical personnel in charge. Furthermore, because workers are sent to elderly patients’ homes alone as a part of HCS, they have to attend to medical emergencies that they are not equipped for, thus explaining the “low” ratings [16]. Moreover, since there are many elderly people in ECF with chronic conditions, they are at risk of dying in the absence of quick and proper treatment, which is possibly why ECF directors reported a lower ability to respond to emergencies than HCC, as their burden is greater owing to the increased potential of permanent damage occurring. 

The focus groups revealed that when the registered nurse was out of the office, or it was night time, it was the caregivers who dealt with emergencies, so they reported a limitation to effective first-aid. In 48.1% of HCC, the caregivers were in charge of first-aid and were relying so much on calling the 119 emergency number that the service warned them about making frequent calls, displaying situational similarity between ECF and HCC. Additionally, when emergencies occurred in ECF and HCC and a medical practitioner was absent, non-medical personnel had to perform medical procedures to save the patients’ lives, leading to conflict about performing tasks that are prohibited due to skill level. Some medical treatments should not be done during certain emergencies, even at the request of a guardian. Thus, there are a variety of difficulties that arise owing to the constraints of the law that differ from what is expected of non-medical personnel, such as different levels of medical practice in different agencies. To provide effective first-aid, clear emergency standards need to be put in place to establish the scope of possible medical practices and the role of non-medical personnel within legal boundaries [6].

Overall, our results showed that both ECF and HCC participants were confused about recognizing emergencies accurately. In some cases, death was confirmed four to five hours later owing to incorrect assessment of the emergency, and some elderly patients were left with unattended fractures and even had to get hip joint surgery. These problems were caused by the late detection of emergencies and worsening patient health. To ensure the health of elderly patients, education needs to be provided to caregivers [17]. Additionally, in order to proactively address the frequent emergencies at ECF and HCC, more than half of the ECF and HCC participants had developed their own regulations or manuals, which they were using. Given the importance of emergency training and the need for continuous education to enable the accurate assessment of emergencies, the use of videos and educational materials can be valuable, as well as receiving training at their district office or fire station. This may be difficult, as educational methods differ between facilities, and training is done using unverified video clips and educational materials. Moreover, almost no education is being provided in HCC. Expressing regret over the lack of compulsory education for emergencies, participants emphasized the need for systematic and professional emergency education to be promoted at the government level. One of the results from the qualitative research that was not found in the quantitative research, was that participants in this study struggle to distinguish where and when a situation should be called an emergency. Additionally, participants in this study felt the burden of legal investigations related to deaths and were frustrated by being questioned at the police station.

Based on these results, it is urgent to prepare a system that can correctly determine and help ECF and HCC respond appropriately to emergencies and cope with emergencies effectively. We propose the following: (1) a manual be prepared along with guidelines on how to assess and respond to emergencies; (2) legal grounds for emergencies that can be applied in long-term care services should be established; (3) systematic, compulsory education led by the government needs to be provided to strengthen the coping capacity of workers; (4) standardization of decision-making by establishing a timeline for initiating an emergency call; and (5) considering an increase in the number of nurses.

## 6. Conclusions

This study examined the type of emergencies occurring in the elderly receiving care services via ECF and HCC and used a mixed-methods design to analyze how effectively care workers responded to these emergencies. Furthermore, qualitative research allowed the participants to clearly and honestly express their feelings about emergencies and their ability to cope with them. According to our results, there is a difference in the status of emergencies and emergency response capacity according to the type of care service. Finally, we hope to develop and apply a training program for each type of care service, utilizing practical guidelines for emergency response management based on the results of this study.

## Figures and Tables

**Table 1 ijerph-17-00066-t001:** Participant characteristics (N = 259).

Classification	Categories	ECF (N = 130)	HCC (N = 129)
		N (%)	N (%)
Long-term care facility type		130	129
Job type of directors	Social worker	110 (84.6)	116 (89.9)
	Nurse	16 (12.3)	8 (6.2)
	Other	4 (3.1)	5 (3.9)
Emergency situation report ^†^	Facility director	44 (33.8)	49 (38.3)
	Nurse	57 (43.8)	13 (10.2)
	Nurse’s aide	18 (13.8)	12 (9.4)
	Guardian	3 (2.3)	25 (19.5)
	Calling 119 rescue team	8 (6.2)	43 (33.6)
In charge of first-aid ^†^	Facility director	23 (17.7)	32 (21.3)
	Nurse	43 (33.1)	12 (8.0)
	Nurse’s aide	36 (27.7)	14 (9.3)
	Caregiver	25 (19.2)	72 (48.1)
	Social worker	3 (2.3)	20 (13.3)
Accompanied during hospital transportation ^†^	Facility director	25 (13.5)	20 (11.0)
	Nurse	24 (13.0)	9 (5.0)
	Nurse’s aide	43 (23.2)	21 (11.6)
	Caregiver	84 (45.4)	86 (47.5)
	Social worker	4 (2.2)	27 (14.9)
	Guardian	5 (2.7)	18 (9.9)
Experience of emergency	Yes	130 (100.0)	129 (100.0)
	No	0 (0.0)	0 (0.0)

^†^ Multiple response. ECF: Elderly care facilities, HCC: Home care center.

**Table 2 ijerph-17-00066-t002:** Emergency situations experienced and first-aid (N = 259).

Characteristics	Emergency Situations Experienced				Characteristics		First-Aid		
	ECF (n = 130)	HCC (n = 129)	t	*p*		ECF (n = 130)	HCC (n = 129)	t	*p*
	M ± SD	M ± SD				M ± SD	M ± SD		
Total	1.87 ± 0.15	0.88 ± 0.25	44.613	<0.001 ***	Total	0.60 ± 0.21	0.42 ± 0.20	6.862	<0.001 ***
Dyspnea	4.00 ± 0.00	1.86 ± 0.72	33.973	<0.001 ***	Vital sign	0.90 ± 0.30	0.68 ± 0.47	4.454	<0.001 ***
Dysphagia	4.00 ± 0.00	2.42 ± 0.69	25.934	<0.001 ***	Ice pack	0.73 ± 0.45	0.57 ± 0.50	2.681	0.008 **
Stomach ache	4.00 ± 0.00	1.21 ± 0.81	39.279	<0.001 ***	Assistance with medication	0.69 ± 0.46	0.62 ± 0.49	1.221	0.223
Psychological symptoms of dementia	4.00 ± 0.00	2.84 ± 0.80	16.301	<0.001 ***	Blood sugar test	0.73 ± 0.45	0.67 ± 0.47	−1.122	0.263
Loss of consciousness	3.42 ± 0.63	1.53 ± 0.71	22.556	<0.001 ***	Insulin injection	0.24 ± 0.43	0.09 ± 0.29	3.176	0.002 **
Hypertension (Hypotension)	2.46 ± 1.06	1.00 ± 0.88	12.121	<0.001 ***	Hypoglycemia	0.65 ± 0.48	0.43 ± 0.50	3.490	0.001 **
Fever	2.29 ± 0.94	0.72 ± 0.70	15.260	<0.001 ***	Oxygen saturation measurement	0.59 ± 0.49	0.12 ± 0.32	9.167	<0.001 ***
Stroke	1.65 ± 0.72	0.60 ± 0.70	11.758	<0.001 ***	Oxygenation	0.69 ± 0.46	0.21 ± 0.41	8.845	<0.001 ***
Dehydration	1.57 ± 0.71	0.33 ± 0.55	15.608	<0.001 ***	Suction	0.62 ± 0.49	0.18 ± 0.38	8.008	<0.001 ***
Heart attack	0.95 ± 0.69	0.16 ± 0.38	11.500	<0.001 ***	Basic life support	0.71 ± 0.46	0.38 ± 0.49	5.587	<0.001 ***
Hypoglycemia	1.25 ± 0.53	1.46 ± 0.70	−2.641	0.009 **	Automated external defibrillators	0.34 ± 0.48	0.16 ± 0.36	3.492	0.001 **
Hematochezia	1.17 ± 0.42	0.24 ± 0.45	17.322	<0.001 ***	Heimlich maneuver	0.68 ± 0.47	0.60 ± 0.49	1.338	0.182
Convulsion	0.67 ± 0.61	0.31 ± 0.51	5.108	<0.001***	Splinting	0.54 ± 0.50	0.36 ± 0.48	2.982	0.003 **
Fracture	1.13 ± 0.34	0.60 ± 0.76	7.151	<0.001 ***	Hemostasis	0.75 ± 0.44	0.46 ± 0.50	4.947	<0.001 ***
Self-mutilation	0.20 ± 0.46	0.10 ± 0.33	2.014	0.045 *	Wound dressing	0.75 ± 0.43	0.46 ± 0.50	5.102	<0.001 ***
Burn	0.49 ± 0.50	0.19 ± 0.39	5.482	<0.001 ***	Convulsion	0.67 ± 0.47	0.44 ± 0.50	3.767	<0.001 ***
Severe bleeding	0.27 ± 0.45	0.16 ± 0.38	2.210	0.028 *	Calling 119	0.85 ± 0.35	0.81 ± 0.40	1.019	0.309
Addiction	0.19 ± 0.40	0.16 ± 0.39	0.604	0.547	Calling guardian	0.92 ± 0.28	0.78 ± 0.42	3.165	0.002 ***

* *p* < 0.05, ** *p* < 0.01, *** *p* < 0.001. ECF: elderly care facilities, HCC: home care center.

**Table 3 ijerph-17-00066-t003:** Workers’ self-reported ability to respond to emergencies (N = 259).

Characteristics	Categories (n)	ECF (N = 130)	HCC (N = 129)	χ^2^	*p*
		N (%)	N (%)		
Emergency response abilities	Very low	32 (24.6)	12 (9.3)	27.115	<0.001 ***
	Low	62 (47.7)	45 (34.9)		
	Average	17 (13.1)	42 (32.6)		
	High	13 (10.0)	26 (20.2)		
	Very high	6 (4.6)	4 (3.1)		
Manual	Yes	71 (54.6)	70 (54.3)	0.003	0.527
	No	59 (45.4)	59 (45.7)		
Emergency room transfer	Yes	128 (98.5)	93 (72.1)	35.961	<0.001 ***
	No	2 (1.5)	36 (27.9)		
Number of emergency room transfer	Less than 5	59 (45.3)	100 (77.5)	47.966	<0.001 ***
	6–10	34 (26.2)	29 (22.5)		
	11–15	13 (10.0)	0 (0.0)		
	16–20	16 (12.3)	0 (0.0)		
	More than 21	8 (6.2)	0 (0.0)		
Decision to transfer to emergency room	Facility director	37 (28.5)	33 (25.6)	43.716	<0.001 ***
	Nurse (Nurse aide)	37 (28.5)	7 (5.4)		
	Caregiver	0 (0.0)	3 (2.3)		
	Guardian	11 (8.5)	43 (33.3)		
	Discuss with guardian	45 (34.6)	43 (33.3)		

*** *p* < 0.001. ECF: elderly care facilities, HCC: home care center.

**Table 4 ijerph-17-00066-t004:** Elderly care facilities and home care services directors’ experiences of emergencies and their ability to respond to emergencies.

Category	Theme Cluster	Theme
Confusion in recognizing emergencies	Ambiguous standards about emergencies	Confusion about identifying an emergency
		Unclear classification of the dying process and emergencies
		Death and emergencies decided by the guardians
	Emergencies that are not recognized accurately	Situation not accurately identified owing to lack of medical expertise
		Caregiver with meager knowledge about emergencies
		Care provider who cannot accurately relay the condition
	Emergencies that are discovered late	Death of the subject that was discovered belatedly
		Worsening of the subject’s health owing to an overlooked emergency
		Elderly living alone for whom emergencies are difficult to identify
Sudden occurrence of an emergency	Emergencies that occur owing to diseases	Frequent hypoglycemic shock
		Emergency due to bleeding in the urethra or uterus
		Obstruction of airway due to sputum
		Problem behavior due to dementia patient’s wandering
	Emergencies related to accidents	Emergencies related to suicide
		Emergencies due to fractures
Unsystematic emergency training	By itself conducted training on emergency response	Effective repetitive training
		Calmly responding to emergencies based on regular monthly training
		Emergency response training during case management
		response using emergency manual at the center
	Basic life support training that can save lives	Emergency training using the national learning center
		Emergency response through video training
		Basic life support training that is helpful during emergencies
	Disappointment about the absence of compulsory education	Disappointment at insufficient emergencies training
		Lack of systematic compulsory education about responding to emergencies
Non-independent response	Response based on consultation with family	Completion of an emergency manual through communication with guardians
		Hospital visitation through consultation with the family
	Conflict with guardian about the emergency	Emergency treatment delayed because of the guardian’s non-arrival
		Guardians that are uncooperative with hospital treatment
		Conflict with a guardian about the performance of medical care
	Emergency responses that are dependent on medical personnel	Concentrated observation for emergency response through takeover training with medical personnel
		Emergency response using the part-time-doctor policy
		Non-independent response following 119 guidelines at room
Legal standards that do not fit reality	Unclear standards of medical care	Unclear standards of medical care at the center
		Limited response due to lack of certification
		Medical care that differs by center
	Laws that are unsuitable in reality	Legally undefined standards of emergency response
		Reality different from legal standards
		Corporation guideline manual is not realistic
	Performing prohibited medical practices	Emergencies in which prohibited medical practice is performed
		Medical care performed at home care center after signing consent
Psychological withdrawal associated with death	Frustration at investigation of death	Feeling burdened about legal investigation of death
		Frustration of having to be questioned by the police owing to death within the center
	Psychological withdrawal that needs to be coped with alone after death	Feelings of guilt after police investigation related to an accident
		Psychological withdrawal after death coped with alone

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
