# Peer review of "Emergencies in Long-Term Care Services for the Elderly in Korea: A Mixed-Methods Study"

_ijerph, 2019, doi:10.3390/ijerph17010066_

Round 1

Reviewer 1 Report

The article proposal addresses a very interesting and very relevant topic. Nevertheless, a more careful reading of the results will allow to more systematically expose the results. The refuge in the set of tests results in the loss of the objective of the work, conditioning the interpretation of the results. Better emergency material and emergency support conditions as technical training seem to be the main reason for the best response in emergency situations.

The restructuring of results reading is needed to answer the article intended objectives. 

First of all, in my opinion, the article to be published should be deeply reformulated. The unbalanced between statistic treatment and the results found is clear and the results analysis must be developed. There is much more information in the tables that are not used, or highlighted.

Second, the survey and the focus group approach must be clarified in terms of what was the objective to use that two approaches and the results analysis must be in line with that objective. A two steps analysis will be much more useful. First the general analysis from the survey and the complement with the focus group results.

In addition, the theme is interesting and needed that type of reflection.

Author Response

"please see the attachment"

Reviewer 2 Report

This topic is of significant importance. I commend the quality of your research student and its potential to influence the safe care of frail older adults.

Suggestions for improvement:

1. Study Objectives:  revise to clearly state the research aims. As written the section repeats the research design which is not needed.

 2. grammar error--line 74 change 'verify' to 'identify'

3. need to communicate a link between the ECF  in Study Participants and LTCS in Study Objectives to justify why these directors were recruited for data in this study.  Providing this link will enable non-healthcare workers to understand the implications of this study.

4. Data Collection: Also add weather or not the participants signed a consent to participate as common criterion for qualitative methods.

5. Grammar issue and confusing:  lines 102-104, Are you trying to relate  the number of different incidents per category the participants reported ?  or what is the meaning of this list?

6. Line 106 --Missing rationale for assigning 4 points to very little experience. Need to add rationale for weighting this variable greater than variables of large numbers of experience.

7. Table on page 4:  '119' having a number as category title is confusing, consider replacing with a descriptive word such as Emergency Call or ambulance. Consider adding description of what is indicated by stating '119'

Line 149:  add clarity,  consider inserting 'self-reported' after Participants'

Line 157 Table title:   consider adding 'self-reported' after Caregivers'

Lines 170-171:  terms 'unsystematic' and 'non-independent' need explanation or context to assist the reader to understand their meaning and significance.

Line 218, average of 0.78 emergencies is a number less than 1. Thus this indicates to the reader that several participants as HCC never had emergency experiences, thus raises concern for the quality of the data and data collection.

line 254:  in addition would you also conclude a need for standards of decision-making by establishing a timeline for initiating an emergency call and also consideration to increase number of nurses

line 261: consider revision to improve readability:  move 'assess emergencies accurately' to be after 'education'

line 278:  clarity of idea, replace 'occurrence' with 'type'.   The word 'occurrence' can imply frequency

Round 2

Reviewer 1 Report

Thank you for the improvement in the present paper.